# Beer or Ethanol Effects on the Body Composition Response to High-Intensity Interval Training. The BEER-HIIT Study

**DOI:** 10.3390/nu11040909

**Published:** 2019-04-23

**Authors:** Cristina Molina-Hidalgo, Alejandro De-la-O, Lucas Jurado-Fasoli, Francisco J. Amaro-Gahete, Manuel J. Castillo

**Affiliations:** EFFECTS 262 Department of Medical Physiology, School of Medicine, University of Granada, 18071 Granada, Spain; delao@ugr.es (A.D.-l.-O); juradofasoli@ugr.es (L.J.-F.); mcgarzon@ugr.es (M.J.C.)

**Keywords:** exercise, alcohol, beer, body composition, training program, high intensity interval training, fat mass, lean mass, visceral adipose tissue

## Abstract

High-intensity interval training (HIIT) is promoted as a time-efficient strategy to improve body composition but concomitant beer intake, which is common among physically active individuals, may interfere with these effects. The primary aim of this study is to determine the effects of a 10-week (2 days/week) HIIT program on anthropometric and body composition measurements, and to assess whether those effects are influenced by the moderate consumption of beer (at least 5 days/week), or its alcohol equivalent. Young (24 ± 6 years old) healthy adults (*n* = 72, 35 females) volunteered for a non-training group (Non-Training group) or for HIIT training. Those going for training choose whether they preferred to receive alcohol or not. Those choosing alcohol were randomly allocated for receiving beer (5.4%; T-Beer group) or the equivalent amount of alcohol (vodka; T-Ethanol group) in sparkling water. Those choosing no-alcohol were randomly allocated for receiving alcohol-free beer (0.0%; T-0.0Beer group) or sparkling water (T-Water group). From Monday through Friday, men ingested 330 mL of the beverage with lunch and 330 mL with dinner; women ingested 330 mL with dinner. Before and after the intervention, anthropometry and body composition, through dual-emission X-ray absorptiometry, were measured. No changes in body mass, waist circumference, waist/hip ratio, visceral adipose tissue or bone mineral density occurred in any of the groups. By contrast, in all the training groups, significant decreases in fat mass together with increases in lean mass (all *p* < 0.05) occurred. These positive effects were not influenced by the regular intake of beer or alcohol. In conclusion, a moderate beer intake does not blunt the positive effect of 10-week HIIT on body composition in young healthy adults.

## 1. Introduction

Physical exercise is an integral component of a healthy life-style, with strong evidence supporting the notion that it can help to lose weight and improve body composition [1]. The international physical activity recommendations provided by the World Health Organization (WHO) include 150 min of aerobic exercise at moderate intensity or 75 min of aerobic exercise at vigorous intensity per week [2]. In addition, strength and power exercises are recommended in order to increase muscle mass or to prevent age associated sarcopenia [3]. However, it has been shown that most adults failed to meet the recommendations of physical activity [1], noting the lack of time as the main barrier to follow an exercise training program [4]. Recently, high-intensity interval training (HIIT), which consists in alternate short bursts of high-intensity exercise and recovery periods, has emerged as an alternative to the traditional exercise recommendations because of its time efficiency [5]. In this sense, HIIT has demonstrated to be effective in the improvement of body composition, by reducing fat mass (FM) [5,6] and increasing muscle mass [7], in normal-weight and in overweight-obese individuals [8]. In fact, there is robust and growing evidence showing that HIIT may elicit greater benefits that moderate-intensity continuous training across a range of health markers [5]. Among them, HIIT has demonstrated to be effective in reducing total, abdominal, and visceral fat mass in both males and females [8]. Similarly, different HIIT protocols have resulted in modest increases in total body and trunk lean mass (LM) [9,10,11].

Beer is a widely consumed beverage in western countries and the most consumed alcoholic beverage in the world [12]. It is consumed by many healthy adults to quench thirst in preference to other beverages particularly after a hard day’s work, as part of social relationships, or after practicing exercise [13,14]. This is particularly the case in a recreational context, where having a beer after a match is considered part of the social aspect of many sport activities [15,16,17]. It has been reported that beer is the most popular alcoholic drink among athletes and sport administrators, with over 90% naming it as their preferred alcoholic beverage [16]. The intake of beer in the exercise recovery phase has been questioned due to its alcohol content [13,18]. Similarly, there is controversy regarding the influence of beer consumption on fat distribution and body composition [19,20,21,22,23,24]. Some authors have reported that moderate beer consumption is not associated with changes in body mass (BM) or body composition parameters [20,22,23,24], while others have reported that alcohol consumption is associated with increased adiposity in adults [19,21]. Moreover, alcohol intake can have influence, not only in FM but also in LM [25]. It has been reported that alcohol can suppress the anabolic response to physical exercise through the reduction of muscle protein synthesis [25].

To the best of our knowledge, there are no previous studies that investigated the influence of a moderate consumption of beer, or alcohol, in the body composition response to a highly demanding physical training precisely designed to induce changes in body composition, in conditions similar to those occurring in real life situations. Therefore, the present study aimed (i) to evaluate the effects of a HIIT program on body composition parameters in healthy adults, and (ii) to analyze whether those effects are influenced by frequent but moderate beer consumption or its alcohol equivalent. We hypothesize that HIIT may improve body composition, but these effects may be blunted by the concomitant intake of beer even in moderate amounts, being these deleterious effects attributed to its alcohol content.

## 2. Materials and Methods

### 2.1. Study Design and Participants

A total of 72 healthy adults (35 females), aged between 18–40, years were enrolled in this registered controlled trial (ClinicalTrials.gov ID: NCT03660579). The study was in accordance with the latest revision of the Helsinki Declaration, and was approved by the Ethics Committee on Human Research at the University of Granada (CEI-Granada) (321-CEIH-2017). The participants were recruited from the province of Granada (Spain) using social networks, local media, and posters. Interested individuals were screened via telephone, and/or e-mail. Several appointments were made in order to provide detailed explanation about the study objectives, design, inclusion criteria, assessments to be undertaken, exercise program intervention, and types of beverages to be ingested. Volunteers were included based on the following inclusion criteria: (i) Body mass index (BMI) from 18.5 to 30 kg/m^2^, (ii) nor engaged in a training program, (iii) having a stable BM during the last 5 months (BM changes <3 kg), (iv) being free of disease, pregnant or lactating women; (iv) not taking any medication for chronic diseases, and (v) no pain, recent lesions or other problems preventing strenuous physical activity. Prior to the study, all participants completed a health exam and gave their written informed consent. All the baseline and follow-up tests were performed at the same setting (Instituto Mixto Universitario Deporte y Salud (IMUDS) at the University of Granada). The total test duration was ~30 min.

As shown in Figure 1, 94 individuals attended an information meeting and were assessed for eligibility. A total of 83 individuals met the inclusion criteria, and, after completing the baseline measurements, they chose whether they preferred to be included in a training (T) or a non-training (N-T) group. Training consisted of two HIIT sessions per week, during 10 consecutive weeks (see below). Those going for training then chose whether they preferred to be included in a group ingesting from Monday to Friday an ethanol containing beverage (5.4% alcohol content) or an alcohol-free beverage. Men ingested 330 mL of that beverage with lunch and 330 mL with dinner. Women ingested 330 mL with dinner. Participants choosing ethanol were randomly allocated either to beer (T-Beer) or to sparkling water with added vodka ethanol (T-Ethanol). Those choosing non-alcohol were randomly allocated either to alcohol-free beer (T-0.0Beer) or to sparkling water (T-Water). Each group was composed of 8 men and 8 women. This type of non-random (based on individual preference) and random allocation of the participants was conducted following ethical considerations and advice made by the ethical committee (321-CEIH-2017). The assessment staff was blinded to the participants’ randomization assignment.

Beverage intake and physical activity levels were registered before and after the intervention program by the Beverage Intake Questionnaire (BEVQ [26]) and self-reported, respectively. In the background questionnaire, participants were asked to report their usual frequency of alcohol intake in seven possible response categories. The lack of adherence to the training protocol (<20% no-shows for training sessions) and/or not complying with the stipulated beverage intake were excluded from the study.

### 2.2. Training Protocol

A 10-week registered controlled trial was conducted. The training sessions took place 2 days/week in the late afternoon or early evening from Monday to Friday, leaving a rest period of, at least, 48 h between training sessions. The training volume was 40–65 min/week following the methodology described by previous studies [27,28,29]. It was divided into two different phases, starting with a familiarization phase to learn the main movement patterns as previous studies have suggested [30,31,32].

The HIIT intensity was programmed considering the updated scientific evidence [28,33,34]. In all cases, the intensity was >8 Rating of Perceived Exertion (0–10 RPE scale) [35], which has a positive linear relationship with heart rate and VO_2max_ [36,37]. The training sessions started with a dynamic standardized warm-up, including several muscle activation exercises (i.e., child’s pose breathing; pelvis bridge; cat camel; upper back rotation; front and side planks; arms Ts; arms Ys; toe walks; high knee walks; walking lunges; side lunges; monster walk; sumo walk and anti-rotational stability press). The participants performed eight weight-bearing exercises in circuit form twice per set (i.e., frontal plank, high knees up, horizontal row, battle rope, squat, dead lift, push up, and burpees). There was a passive rest between exercise, and an active rest between sets (an intensity of 6 RPE, which corresponds with 60% VO_2max_ [36,38]), following the periodization described in previous studies [30]. The training sessions ended with a cooling-down protocol (active global stretching) including anterior and posterior chain exercises (i.e., pigeon pose; lying twist; figure four stretch; lunging hip flexor stretch; biceps stretch, and trapezius neck stretch).

### 2.3. Beverage Intake Protocol

The beverages were ingested daily from Monday to Friday. The volumes of fluid ingested were the same in all groups (660 mL for men and 330 mL for women), men ingested 330 mL with lunch and 330 mL with dinner, women ingested 330 mL with dinner: (i) T-Beer group ingested regular Lager Beer (5.4% alcohol-Alhambra Especial^®^, Granada, Spain); (ii) T-0.0Beer group ingested alcohol-free beer (0.0% alcohol-Cruzcampo^®^, Sevilla, Spain); (iii) T-Water group ingested sparkling water (Eliqua 2^®^, Font Salem, Spain); (iv) T-Ethanol group ingested sparkling water with exactly the same amount of distilled alcohol added. The distilled alcoholic beverage used in our study was branded vodka because of the purity of its composition (37.5% ethanol and 62.5% water). We based on scientific evidence to select the amount of alcohol ingested by the participants, which defines a moderate amount as two or three drinks/day or 24–36 g of ethanol/day for men and one to two drinks/day or 12–24 g of ethanol/day for women [23,39]. The beverages were coded and provided by a staff member of our research laboratory at the beginning of each week. The investigators who did the evaluations were not aware of the group assignment of the participant. The participants included in alcohol groups were strictly instructed to drink a moderate amount of alcohol during the weekend (i.e., 660 mL/day for men and 330 mL/day for women). Those included in non-alcohol groups were requested to refrain from alcohol also during the weekend.

### 2.4. Anthropometric Parameters and Body Composition Assessment

Anthropometric parameters and body composition assessment were conducted before and after the 10-week intervention program. BM and height were measured without shoes and with light clothing, using a pre-validated scale and stadiometer (model 799, Electronic Column Scale, Hamburg, Germany), and the body mass index was calculated (weight/height^2^) [40].

Waist circumference (WC) and hip circumference (HC) were measured in triplicate with a non-elastic tape (Seca 200, MWS Ltd., Scalesmart, Hamburg, Germany) to the nearest 0.1 cm and the mean of the 3 measurements was used in the analyses. WC was measured at the mid-point between the bottom of the rib cage and the iliac crest at the end of a normal expiration [41]. HC was measured around the widest portion of the buttocks [41]. Waist-hip ratio (WHiR) was calculated by dividing waist by hip values.

FM, visceral adipose tissue (VAT), LM, and bone mineral density (BMD) were measured by a dual-energy X-ray absorptiometry (DXA) scanner (Discovery Wi, Hologic, Inc., Bedford, MA, USA). The whole-body scan was used to obtain all body composition parameters. We conducted the quality controls, the positioning of the participants, and the analyses of the results following the manufacturer’s recommendations. An automatic delineation of the anatomic regions was performed by the software APEX 4.0.2. FM was also expressed as percentage of body mass.

### 2.5. Statistical Analysis

Sample size calculations were based on a minimum predicted 15% change in BMI, FM, and LM between the intervention groups and the non-training group, with an expected standard deviation of 15%. A sample size of 13 participants was predicted to provide a statistical power of 85% considering a type I error of 0.05 [42], based on a pilot study. However, we recruited a minimum of 16 participants per group (a total of 80) to accommodate for a maximum loss of 20% at follow-up.

Data were checked for normality with use of a Shapiro–Wilk test and visual inspection of Q-Q plots. We conducted repeated-measures analysis of variance in order to study changes in BM, BMI, WC, HC, WHiR, FM, FM percentage, VAT, LM, and BMD across time, between groups, and the interaction (time × group). An analysis of covariance (ANCOVA) was conducted to determine the effect of the groups (fixed factor) on anthropometric and body composition outcomes, i.e., post-FM minus pre-FM (dependent variable), adjusting for the baseline values (model 1). We conducted the same analysis for changes in WC, FM percentage, VAT, LM, and BMD. Bonferroni post hoc tests with adjustment for multiple comparisons were also performed. Additional models were conducted controlling for baseline values and sex (model 2), and for baseline values and age (model 3) (see Appendix A). The level of statistical significance was defined at *p* < 0.05. As no significant interaction was obtained by sex, we fitted all models including men and women together. The statistical analyses were conducted in Statistical Package for Social Science (SPSS, V. 25.0, IBM SPSS Statistics, IBM Corporation), and the graphical plots were conducted in GraphPad Prism 5 (GraphPad Software, San Diego, CA, USA).

## 3. Results

The participants flow-chart is presented in Figure 1. During the study period, a total of 11 participants (6 men and 5 women) dropped out due to insufficient attendance in the training sessions or the difficulty to fulfill the protocol requirements. The baseline characteristics of the study participants can be observed in Table 1. There were no differences among groups at the baseline. The reported intakes of alcohol in the different groups were also similar (*p* = 0.144; See Table 1). The distribution in the number of men and women was nearly equal in each group.

Table 1 shows BM, BMI, WC, HC, and WHiR before and after the intervention study. Repeated measures analysis of variance (ANOVA) revealed no effect of training neither of type of beverage in BM (*p* = 0.849), BMI (*p* = 0.842), and HC (*p* = 0.900), while a significant effect of training and of the beverage type was observed in WHiR and HC (*p* = 0.029, and *p* = 0.003, respectively).

Figure 2 shows changes in FM, WC, and LM after the intervention study among the five groups. ANCOVA revealed no significant differences among groups in FM (*p* = 0.156; Figure 2A), whereas significant differences were noted in WC and LM (*p* = 0.007, and *p* ≤ 0.001, respectively; Figure 2B,C). The results remained after controlling by sex and age in FM (all *p* > 0.05; see Appendix A), also in WC and LM (all *p* < 0.05; see Appendix A). A significantly higher LM post-intervention was noted in all training groups compared with the non-training group (T-Beer, *p* ≤ 0.001; T-0.0Beer, *p* = 0.005; T-Water, *p* ≤ 0.001; and T-Ethanol, *p* ≤ 0.001).

Figure 2 also shows changes in FM percentage, VAT, and BMD after the intervention study among the five groups. ANCOVA also revealed significant differences between groups in FM percentage (*p* = 0.009; Figure 2D), whereas no significant differences were noted neither in VAT nor in BMD (*p* = 0.627, and *p* = 0.474, respectively; Figure 2E,F). The results remained after controlling by sex in FM percentage (all *p* ≤ 0.013; see Appendix A), also in VAT and BMD (all *p* > 0.05; see Appendix A). A significantly lower FM percentage post-intervention was noted in T-Beer, T-Water, and T-Ethanol compared with the non-training group (all *p* < 0.05; Figure 2D).

## 4. Discussion

The primary findings of our study are that 10 weeks of HIIT did not have an influence on BM, but this type of training significantly decreased FM and FM percentage and increased LM in healthy adults. These positive effects were not affected by the concomitant regular intake of beer, or its alcohol equivalent, in moderate amounts. Neither HIIT nor beer or alcohol intake influenced adipose tissue distribution or BMD. The lack of effect on BM or BMI was the result of the simultaneous decrease in FM and increase in LM.

The role of a HIIT program on body composition parameters have been analyzed previously in several studies [43]. Some recent meta-analysis have reported that different HIIT protocols did no modify BMI and WC in normal-weight individuals [6,44], which concurred with the findings of this study. In addition, current systematic and meta-analysis reviews have concluded that a HIIT program appears to be effective on FM reduction (~6%) [5,6,8], and can alter VAT showing a larger decrease compared to caloric restriction (−6.1%) [45]. Our results agree in this line, since HIIT showed a significant decrease of FM and FM percentage (all *p* < 0.05) in T-0.0Beer and T-Water groups, also a significant decrease of VAT (all *p* < 0.05); whereas no significant changes were found on BMI or WHiR (all *p* > 0.05) after the HIIT intervention in any group (see Table 1). Moreover, it has been previously reported that HIIT is a time efficient and effective method to stimulate muscle size adaptations in individuals with a BMI between 25–45 kg/m^2^ after 3 sessions/week (14% increase in muscle cross-sectional area) during the 3-week intervention [7], and in non-obese young adult women after 3 sessions/week (5–6% increase in LM) during the 12-week HIIT intervention [6]. These results agree with our findings, since a significant increase of LM (+5%) were obtained in T-0.0Beer and T-Water groups (see Table 1). Our results are in accord with a previous study by Nybo et al. [46] who found no significant changes in BMD after a 12-week HIIT intervention. However, exercise programs that combine high impact activity with resistance training has been shown to be the most effective in augmenting BMD [47,48]. The low number of cases in our groups may be the cause of not finding a statistically significant effect.

The effects of alcohol consumption on body composition have been debatable. The relationship between alcohol consumption and central adiposity has not been clearly or consistently reported in the literature with some studies reporting a positive association between alcohol consumption and WC or WHiR [21,49,50], while others have reported an inverse relationship [24,51]. A previous study observed that alcohol ingestion was not associated with BMI, WHiR, and WC in adults aged between 35 and 64 years in a daily low-to-moderate alcohol consumption [21]. In fact, a cross-sectional study found an inversely association between drinking frequency and BMI, where the lowest odds of being obese was observed among the most frequent drinkers [24]. Our results agree with previous findings, since no group showed changes in BMI, WHiR, and WC, independently of the beverage ingested during the intervention. Some studies have reported that alcohol consumption is associated with an increase in VAT [21,49,50], and could stimulate lipogenesis and inhibit lipolysis on healthy adults [21]. In this sense, it is believed that beer consumption is associated with increased WC or WHiR, particularly in men, a phenomenon popularly referred to as “beer belly” [52]. This belief might be supported by cross-sectional research, reporting abdominal obesity as being associated with beer consumption [53]. However, some prospective studies have shown inconsistent results, such as the study of Schütze et al. [52], who have reported only limited evidence for a site-specific effect of beer drinking on WC. Notwithstanding, the effect of alcohol on fat metabolism remains obscure, our results agree with those obtained by Kim et al. [50], who found that the participants decreased in subcutaneous adipose tissue in spite of their alcohol intake. Further, our results in FM concur with those obtained by Brandhagen et al. [19], since all participants including alcohol consumption groups, decreased in FM percentage (see Figure 2A,D). In addition, all interventions groups, T-Beer and T-0.0Beer groups included, showed no negative changes in VAT, WC or WHiR. Furthermore, T-0.0Beer group showed a significant decrease in VAT and WC, and a clear trend decrease in WHiR, while T-Beer and T-Ethanol groups did not show impairment in either of variables (see Table 1).

On the other hand, it seems clear that alcohol consumption could reduce muscle protein synthesis, suppressing the anabolic response in skeletal muscle [25]. In this line, Coulson et al. [49] found that those groups consuming three or more alcoholic drinks on usual drinking days had lower lean mass than non-drinkers, and this association was not attenuated by adjustment for physical activity levels. Thus, it could be expected that the positive effects of an exercise program on LM could be attenuated by alcohol consumption. However, our results do not agree with these previous findings, since T-Beer and T-Ethanol groups increased their LM (+5%; see Table 1), improving similarly in all interventions groups independently of the type of beverage intake. Therefore, the moderate consumption of alcohol did not seem to influence the anabolic response of a HIIT program of 10-weeks. Our results agree with the data obtained by Viena et al. [54] in a recent meta-analysis, who concluded that HIIT is a useful tool to reduce FM and FM percentage. In our study, all intervention groups showed a reduction in FM and FM percentage after a 10-weeks HIIT program, and these positive effects were not influenced by the concomitant regular intake of beer, or its alcohol equivalent, in moderate amounts. Apart from that, while the evidence regarding the impact of alcohol use on BMD is inconclusive, longitudinal studies have shown that exercise with high-impact load may provide an effective osteogenic stimulus [46]. However, in this study no changes were found in BMD neither in T-Beer and T-Ethanol groups nor T-0.0Beer and T-Water groups. The lack of changes in BMD in any group could be due to the short duration of the intervention program, since longer programs are needed to induce improvements in BMD [55,56].

The present study has some limitations. Firstly, the sample size was relatively small to study the influence of different alcohol beverages in moderate amounts during an exercise training intervention on body composition considering both sexes separately, although no interaction effects were observed. Considering that we compared a total of four different types of beverages ingested, our study could be underpowered to note statistical differences in specific body composition-related parameters between them. Therefore, further studies are needed to clarify the long-term effects (>10 weeks) of HIIT and other training modalities with the ingestion of a moderate dose of alcohol on body composition parameters. Secondly, physical activity was not monitored with tri-axial accelerometer for movement registration. Although we did not collect dietary data during the intervention, we assessed the adherence to the Mediterranean diet by PREDIMED questionnaire before the intervention, finding that our sample did not adhere to the Mediterranean diet. Therefore, our results could be extrapolated to other dietary patterns. Further studies are needed to clarify what is the role of the dietary pattern on body composition parameters during an exercise program intervention combined with moderate alcohol consumption. Finally, participants were not purely randomized, they were asked to choose their preferences about being included in a training or in a non-training group, or in an alcohol or an alcohol-free group, basically due to ethical considerations, but also reflecting the common reality of the daily life.

## 5. Conclusions

In conclusion, our results show that, in healthy adults, a 10-week HIIT program improves body composition by decreasing FM and increasing LM, and these positive effects are not influenced by the concomitant intake of beer, or its alcohol equivalent, in moderate amounts. In addition, the intake of beer, or its alcohol equivalent, while exercising does not affect body fat distribution.

## Figures and Tables

**Figure 1 nutrients-11-00909-f001:**
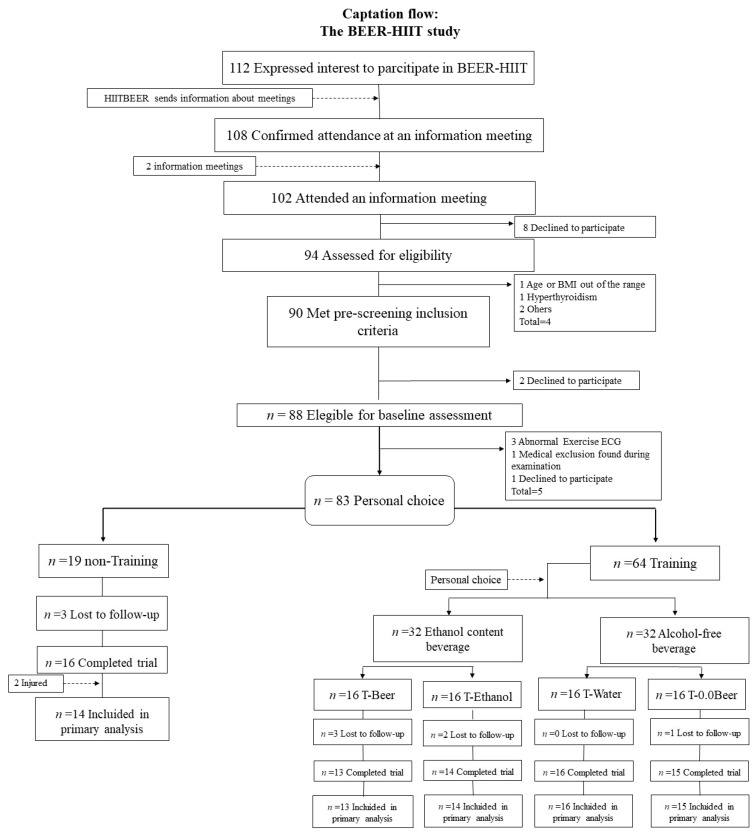
Flow-chart diagram. Abbreviations: BMI, body mass index; ECG, electrocardiogram; N-T, non-training group; T-Beer, group that performed HIIT and consumed alcohol beer; T-0.0Beer, group that performed HIIT and consumed non-alcoholic beer; T-Water, group that performed HIIT and consumed sparkling water; T-Ethanol, group that performed HIIT and consumed sparkling water with alcohol added.

**Figure 2 nutrients-11-00909-f002:**
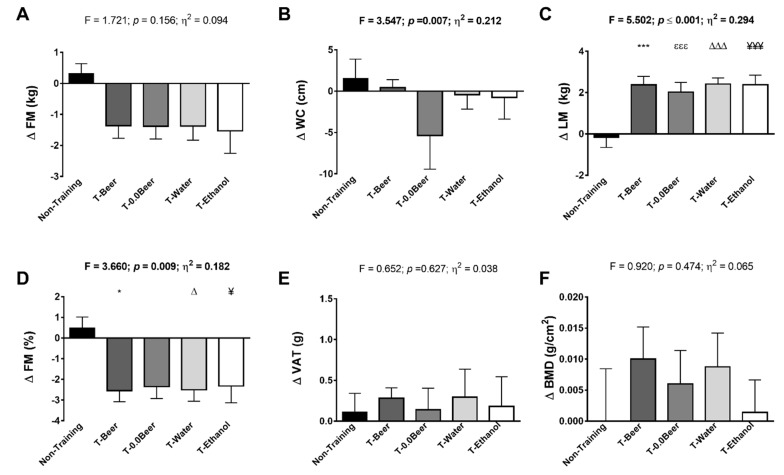
Changes in (**A**): fat mass (FM), (**B**): waist circumference (**WC**), (**C**): lean mass (**LM**), (**D**): fat mass (**FM**), (**E**): visceral adipose tissue (**VAT**), and (**F**): bone mineral density (BMD) after the intervention study among the five groups. Significant differences between groups applying an analysis of covariance adjusting by baseline values, with post hoc Bonferroni-corrected t-test, are indicated as: Non-training group vs. T-Beer (* *p* < 0.05), (*** *p* < 0.001); Non-training group vs. T-0.0Beer (εεε *p* < 0.01); Non-training group vs. T-Water (∆ *p* < 0.05), (∆∆∆ *p* < 0.001); and Non-training group vs. T-Ethanol (¥ *p* < 0,05), (¥¥¥ *p* ≤ 0.001). Data are shown as means ± standard error of the mean. Abbreviations: FM, fat mass (kg); WC, waist circumference; LM, lean mass; FM %, fat mass percentage; VAT, visceral adipose tissue; BMD, bone mineral density; N-T, non-training group; T-Beer, group that performed HIIT and ingested alcohol beer; T-0.0Beer, group that performed HIIT and ingested non-alcoholic beer; T-Water, group that performed HIIT and ingested sparkling water; T-Ethanol, group that performed HIIT and ingested sparkling water with alcohol added.

**Table 1 nutrients-11-00909-t001:** Descriptive parameters and anthropometric and body composition variables before and after the intervention program.

	**Non-Training** **(*n* = 14)**				**T-Beer** **(*n* = 13)**				**T-0.0Beer ** **(*n* = 15)**				**T-Water** **(*n* = 16)**				**T-Ethanol ** **(*n* = 14)**				
Age (years)	20.1 (2.4)				24.5 (5.6)				24.5 (5.6)				24.6 (6.6)				26.1 (6.7)				
Sex (%)																					
Men	7 (50.0)				6 (46.2)				8 (53.3)				9 (56.3)				7 (50.0)				
Women	7 (50.0)				7 (53.8)				7 (46.7)				7 (43.8)				7 (50.0)				
*Beverage intake questionnaire (BVQ)*																					
Weekly Alcohol Intake	5.9 (4.4)				3.2 (3.5)				9.3 (8.1)				8.0 (8.2)				5.3 (7.1)				
**Test of Change over Time**																					**Test of Treatment Effects**
	**PRE**	**POST**	**%**	***p***	**PRE**	**POST**	**%**	***p***	**PRE**	**POST**	**%**	***p***	**PRE**	**POST**	**%**	***p***	**PRE**	**POST**	**%**	***p***	***p*** **Time*Group**
*Anthropometric variables*																					
Body mass	64.7 (10.8)	65.0 (11.5)	9.4	0.294	70.1 (15.6)	70.9 (15.8)	1.2	**0.050**	71.5 (16.9)	71.9 (15.8)	0.6	0.302	69.6 (10.1)	70.5 (10.8)	1.3	0.087	68.3 (14.4)	68.8 (14.1)	0.7	0.311	0.849
BMI (kg/m^2^)	22.1 (2.0)	22.2 (2.1)	0.5	0.335	24.0 (4.3)	24.3 (4.3)	1.2	**0.048**	24.99 (3.7)	25.13 (3.8)	0.6	0.279	24.86 (3.4)	25.15 (3.5)	1.2	0.088	24.22 (4.2)	24.40 (4.0)	0.7	0.298	0.842
WC	74.1 (0.1)	75.6 (6.0)	2.0	0.143	80.1 (11.9)	80.5 (11.9)	0.5	0.668	85.8 (11.5)	80.5 (9.9)	−6.2	**0.014**	82.2 (10.1)	81.8 (10.0)	−0.5	0.627	83.1 (10.1)	81.8 (10.0)	−1.6	0.547	**0.003**
HC	95.4 (5.3)	95.6 (6.0)	0.2	0.846	97.0 (9.8)	98.1 (9.5)	1.1	0.338	98.8 (10.1)	100.2 (7.4)	1.4	0.444	99.4 (6.9)	100.2 (6.1)	1.4	0.373	98.6 (6.9)	98.5 (6.8)	−0.1	0.945	0.900
WHiR	0.7 (0.1)	0.8 (0.0)	2.1	0.189	0.8 (0.1)	0.7 (0.1)	−0.7	0.529	0.9 (0.2)	0.8 (0.1)	−8.6	0.063	0.8 (0.1)	0.7 (0.1)	−1.2	0.306	0.8 (0.11)	0.7 (0.09)	−0.8	0.704	**0.029**
*Body composition variables*																					
FM (kg)	17.0 (5.9)	17.3 (5.7)	1.9	0.327	20.0 (7.4)	18.7 (7.3)	−6.8	**0.005**	22.4 (5.7)	20.9 (5.9)	−6.2	**0.004**	22.1 (7.0)	20.8 (6.7)	−6.2	**0.009**	19.9 (7.1)	18.4 (5.5)	−7.7	0.056	**0.048**
FM (%)	26.93 (8.9)	27.42 (8.7)	1.8	0.378	29.0 (7.6)	26.4 (7.2)	−8.8	**<0.001**	32.2 (5.4)	29.8 (5.6)	−7.3	**0.001**	32.3 (7.7)	29.8 (6.9)	−7.7	**<0.001**	29.7 (8.0)	27.4 (7.2)	−7.8	**0.014**	**0.003**
VAT (g)	212.3 (69.6)	213.5 (91.3)	0.5	0.908	282.5 (161.2)	254.5 (162.8)	−9.9	0.094	320.1 (154.4)	280.8 (174.3)	−12.3	**0.015**	321.4 (170.5)	283.2 (169.6)	−11.9	**0.004**	292.8 (189.5)	262.8 (145.0)	−10.2	0.223	0.342

Values of main variables before and after the 10-week intervention program for the non-training and different training groups. Change scores are mean ± standard deviation. Reported p-values are for tests of an overall change over time among all subjects, and for tests of a treatment effect between different intervention groups over time. * Boldface values indicate significance differences (*p* < 0.05). Only participants with complete intervention program were included in the complete analysis. Abbreviations: BMI, body mass index; WC, waist circumference, HR, hip circumference; WHiR, waist-hip ratio; FM, fat mass (kg); FM %, fat mass percentage; VAT, visceral adipose tissue; T-Beer, group that performed HIIT and consumed alcohol beer; T-0.0Beer, group that performed HIIT and consumed non-alcoholic beer; T-Water, group that performed HIIT and consumed sparkling water; T-Ethanol, group that performed HIIT and consumed sparkling water with alcohol added.

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
