# Peer review of "Beer or Ethanol Effects on the Body Composition Response to High-Intensity Interval Training. The BEER-HIIT Study"

_nutrients, 2019, doi:10.3390/nu11040909_

Reviewer 1 Report

Please see the following comments:

Throughout manuscript please replace “body weight” with “body mass” to reflect what you are measuring.

In Methods (Ln 87):  what do you mean by having a stable body mass?  What was the allowable change +/-?

Did you ask the participants how many alcoholic drinks they had during a normal week before the study?

What was their blood alcohol level (approx.) after their drinks?  Why wasn’t the amount of alcohol normalized by subject’s body mass?  Why wasn’t females alcohol intake not split up like the males?  Was the alcohol intake taken right before the workout during then evening?  How could you compare males to females in your study?

In your methods, please state the approx. time the testing occurred?

Even though you mentioned in your limitations that you did not randomize your participants in your study. How this could affected your data?  Bias?  This needs to be stated when discussing your results in the discussion. 

On Ln 334-335, “Secondly, we just control the levels of physical activity and beverage intake before and after the intervention program, not during the study period.”  Is this what you meant to say, why didn’t you control exercise and alcohol during the study.

Why didn’t you control the diet, it makes it hard to make definitive statements.  This is a major confounding variable.  Please discuss further in your discussion. 

Author Response

Comment 1: Throughout manuscript please replace “body weight” with “body mass” to reflect what you are measuring.

Authors’ reply 1: Done, thank you very much.

Comment 2: In Methods (Ln 87):  what do you mean by having a stable body mass?  What was the allowable change +/-?

Authors’ reply 2: We thank the reviewer’s comment. We considered body mass changes < 3kg as “having a stable body weight”. We have clarified this issue in the Methods section (see Ln 86).

Comment 3: Did you ask the participants how many alcoholic drinks they had during a normal week before the study?

Authors’ reply 3: We appreciate the Reviewer’s comment. The number of alcohol drinks was controlled by the Beverage Intake Questionnaire (BEVQ). This questionnaire was developed to estimate mean daily intake of water, sugar-sweetened beverages, and alcohol beverages. To score the BEVQ, frequency (“How often”) is converted to the unit of times per day, then multiplied by the amount consumed (“How much each time”) to provide average daily and weekly intake beverage consumption. To quantify total alcohol beverage consumption, beverage categories containing added alcohol were summed (alcohol beer, wine, spirits, and cocktails). We conducted a sensivity analysis including changes in total alcohol beverage consumption as a covariate, and the results persisted in all cases. 

Comment 4: What was their blood alcohol level (approx.) after their drinks?  Why wasn’t the amount of alcohol normalized by subject’s body mass?  Why wasn’t female’s alcohol intake not split up like the males?  Was the alcohol intake taken right before the workout during then evening?  How could you compare males to females in your study?

Authors’ reply 4: We appreciate the Reviewer’s comment.

We did not measure blood alcohol levels in this study. In fact the drink was provided to the participants and they drink it at home with the corresponding meal. In a preliminary study we measured breath alcohol after ingesting the same amount of alcohol ingested with a standard breakfast. In those circumstances, the blood alcohol levels were approximately 0.24 g/L for men and 0.28 g/L for women.

In our study we tried to mirror as much as possible a real life situation our social context, where the amounts of beer consumed is standard (bottle or can). The amount of alcohol by subject’s body mass was not normalized because we based on scientific evidence to select the amount of alcohol ingested by the participants, which define a moderate amount as two or three drinks/day or 24-36 grams of ethanol/day for men and one to two drinks/day or 12-24 grams of ethanol/day for women [1],[2].

The alcohol intake by the females was not split up like the males because their alcohol amount was significantly less. Given that it is socially accepted that both men and women take a beer (~330 ml) just after a training session, we wanted to simulate this situation.

The alcohol intake was consumed either during the lunch (~3 hours before the training) in men or one hour after the training session in men and women. So it was not ingested immediately before the work-out

We decided to study men and women together based on that no significant interaction was obtained by sex when the statistical analyses were conducted (see statistical section).

Comment 5: In your methods, please state the approx. time the testing occurred?

Authors’ reply 5: Done, thank you very much (see Ln 90-92).

Comment 6: Even though you mentioned in your limitations that you did not randomize your participants in your study. How this could have affected your data?  Bias?  This needs to be stated when discussing your results in the discussion.

Authors’ reply 6: We appreciate the Reviewer’s comment. We did not perform a purely random allocation based on ethical reasons. Following the advice of the ethical committee, is does not seem appropriate to force individuals to drink or not alcohol if he/she does not desire or have the habitude. We have mentioned this issue in the limitations section, since we do not know how this decision could affect our data. In addition, as previously stated we intended to mirror real life situations where training or drinking beer with meals is a personal choice influenced by personal and social factors.

Comment 7: On Ln 334-335, “Secondly, we just control the levels of physical activity and beverage intake before and after the intervention program, not during the study period.”  Is this what you meant to say, why didn’t you control exercise and alcohol during the study.

Authors’ reply 7: We thank the Reviewer’s comment. We did not objectively control (i.e. accelerometry) physical activity levels during the exercise intervention. However, the participants of the current study were strictly instructed to maintain their physical activity and nutritional habits during the intervention. In every training session, the participants received reinforcement advice on not changing their dietary habits except for the intake of the beverage provided. In addition, we have also explained which is considered a moderately intake of alcohol during the weekend in order to ensure the participants adherence to the pre-defined conditions (see Ln 147-150). We have clarified this sentence in the limitations section.

Comment 8: Why didn’t you control the diet; it makes it hard to make definitive statements.  This is a major confounding variable.  Please discuss further in your discussion.

Authors’ reply 8: We understand the Reviewer´s comment. The lack of dietary control could bias the results obtained in our study, which is a limitation of our study (see limitations section). However, the participants were rigorously instructed to maintain their dietary habits during the intervention. Although we did not perform an objective assessment of dietary parameters, we asked to the participants after the intervention whether they had modified their lifestyle habits during the study and no changes were self-reported.

References:

1.        Poli, A. et al. Moderate alcohol use and health: A consensus document. Nutr. Metab. Cardiovasc. Dis. 23, 487–504 (2013).https://doi.org/10.1016/j.numecd.2013.02.007

2.        Meister, K. A., Whelan, E. M. & Kava, R. The Health Effects of Moderate Alcohol Intake in Humans: An Epidemiologic Review. Crit. Rev. Clin. Lab. Sci. 37, 261–296 (2000).https://doi.org/10.1080/10408360091174222

Reviewer 2 Report

General Comments:

This is an interesting topic: as the authors note, the 10-weeks HIIT program improves body composition, however, these positive effects are not influenced by the concomitant intake of beer or its alcohol equivalent in a moderate amount. But I have some reservations about the adequacy of some limitation such as participants were not purely randomized during the design of the study. Additionally, the participants choose self their preferences about being included in training or in a non-training group. However, the approach of study is highly appropriate, although the authors overstate the importance of their study in discussion. The use of statistics seems appropriate.

Review Comments:

- I think the title should be more specific. I suggest giving up the subtitle “The BEER-HIIT Study”.

- Line 16 to 17: The clarified description about the subjects should be mentioned in the abstract. Also, the description of the study groups (Non-Training, T-Beer, T-0.0 Beer, T-Water, and T-Ethanol) 

- Line 66: Are you sure no studies investigated the effects of studied variables on body composition? 

- Line 146 to 147: How Saturday and Sunday days were free of any recommendation?

- Line 147 to 148: The authors claim, if the participants drink alcohol during the Saturday and Sunday, they instructed to do it with a moderate amount. How authors control this investigated independent variable during the experiment period?      

- Line 342: WHY the participants were not purely randomized?

- Line 343: WHY the participants prefer to be included in training or in a non-training group?

- The sentences length is too long in discussion. Please rephrase the length of sentences again.

- Please revise the conclusion in the abstract

Author Response

#Reviewer 2:

Comment 1: I think the title should be more specific. I suggest giving up the subtitle “The BEER-HIIT Study”.

Authors’ reply 1: We think that to be more specific in the title will lengthens it considerably.

Regarding the acronym, since there are other papers in preparation with important physiological and psychological aspects of the study we feel that the use of the acronym is convenient. We leave in the hands of the editor the decision of suppressing or not the acronym from the title and we agree with his/her decision whatever it will be.

Comment 2: Line 16 to 17: The clarified description about the subjects should be mentioned in the abstract. Also, the description of the study groups (Non-Training, T-Beer, T-0.0 Beer, T-Water, and T-Ethanol).

Authors’ reply 2: We appreciate the Reviewer’s comment. Done.

Comment 3: Line 66: Are you sure no studies investigated the effects of studied variables on body composition?

Authors’ reply 3: We have found some studies investigating the effect of HIIT on body composition outcomes [1]–[5]. Similarly, there are different studies that have investigated the effect of beer or alcohol consumption on body composition outcomes [6]–[11]. However, to the best of our knowledge, there are no studies investigating the effects of HIIT and alcohol consumption simultaneously on body composition.

Comment 4: Line 146 to 147: How Saturday and Sunday days were free of any recommendation?

Authors’ reply 4: We understand the Reviewer’s comment. The Saturday and Sunday days were not free of specific recommendation, since the participants allocated in alcohol groups were strictly instructed to consume a moderate intake of alcohol during the weekend. To clarify this issue, we have modified this sentence.

Comment 5: Line 147 to 148: The authors claim, if the participants drink alcohol during the Saturday and Sunday, they instructed to do it with a moderate amount. How authors control this investigated independent variable during the experiment period?     

Authors’ reply 5: We appreciate the Reviewer´s comment. We have included a specific sentence about what is a moderate intake of alcohol during the weekend. The participants were systematically asked about their alcohol consumption during the intervention ensuring that they have met the pre-defined criteria. In addition, our aim when performing this study was to mirror as much as possible the real life situation in our social context.

Comment 6: Line 342: WHY the participants were not purely randomized?

Authors’ reply 6:  We appreciate the Reviewer’s comment. We did not perform a purely random allocation based on ethical reasons. Following the advice of the ethical committee, is does not seem appropriate to force individuals to drink or not alcohol if he/she does not desire or have the habitude. We have mentioned this issue in the limitations section, since we do not know how this decision could affect our data. Similarly, training or not is a personal choice and this type of training is highly demanding. Participants in the control group were instructed on the convenience of an active life-style but they should not be engaged in any training program during the duration of the study.

Comment 7: The sentences length is too long in discussion. Please rephrase the length of sentences again.

Authors’ reply 7: Done, thank you very much.

Comment 8: Please revise the conclusion in the abstract

Authors’ reply 8: We appreciate the Reviewer´s comment. We have slightly modified these conclusions. 

References:

1.        Wewege, M., van den Berg, R., Ward, R. E. & Keech, A. The effects of high-intensity interval training vs. moderate-intensity continuous training on body composition in overweight and obese adults: a systematic review and meta-analysis. Obes. Rev. 18, 635–646 (2017).https://doi.org/10.1111/obr.12532

2.        Brown, E. C., Hew-Butler, T., Marks, C. R. C., Butcher, S. J. & Choi, M. D. The Impact of Different High-Intensity Interval Training Protocols on Body Composition and Physical Fitness in Healthy Young Adult Females. Biores. Open Access 7, 177–185 (2018).https://doi.org/10.1089/biores.2018.0032

3.        Maillard, F., Pereira, B. & Boisseau, N. Effect of High-Intensity Interval Training on Total, Abdominal and Visceral Fat Mass: A Meta-Analysis. Sport. Med. 48, 269–288 (2018).https://doi.org/10.1007/s40279-017-0807-y

4.        Verheggen, R. J. H. M. et al. A systematic review and meta-analysis on the effects of exercise training versus hypocaloric diet: distinct effects on body weight and visceral adipose tissue. Obes. Rev. 17, 664–690 (2016).https://doi.org/10.1111/obr.12406

5.        Blue, M. N. M., Smith-Ryan, A. E., Trexler, E. T. & Hirsch, K. R. The effects of high intensity interval training on muscle size and quality in overweight and obese adults. J. Sci. Med. Sport 21, 207–212 (2018).https://doi.org/10.1016/j.jsams.2017.06.001

6.        Da Rocha, T. F., Hasselmann, M. H., Chaves Curioni, C., Bezerra, F. F. & Faerstein, E. Alcohol consumption is associated with DXA measurement of adiposity: the Pró-Saúde Study, Brazil. Eur. J. Nutr. 56, 1983–1991 (2017).https://doi.org/10.1007/s00394-016-1240-y

7.        Coulson, C. E., Williams, L. J., Brennan, S. L. & Pasco, J. A. Alcohol consumption and body composition in a population-based sample of elderly Australian men. 183–192 (2013).https://doi.org/10.1007/s40520-013-0026-9

8.        Kim, K. H. et al. Alcohol consumption and its relation to visceral and subcutaneous adipose tissues in healthy male Koreans. Ann. Nutr. Metab. 60, 52–61 (2012).https://doi.org/10.1159/000334710

9.        Tolstrup, J. S. et al. The relation between drinking pattern and body mass index and waist and hip circumference. Int. J. Obes. 29, 490–497 (2005).https://doi.org/10.1038/sj.ijo.0802874

10.      Wang, L.; Lee, I-M.; Manson, J. E.; Buring, J. E.; Sesso, H. D. Alcohol Consumption, Weight Gain, and Risk of Becoming Overweight in Middle-aged and Older Women. Arch. Intern. Med. 170, 453–461 (2010).https://doi.org/10.3945/ajcn.115.118406.

11.      Schütze, M. et al. Beer consumption and the ‘beer belly’: scientific basis or common belief? Eur. J. Clin. Nutr. 63, 1143–1149 (2009).https://doi.org/10.1038/ejcn.2009.39